# An Environmental Health Typology as a Contributor to Sustainable Regional Urban Planning: The Case of the Metropolitan Region of São Paulo (MRSP)

**Natasha Ceretti Maria [1],\*** , **Antônio Ralph Medeiros-Sousa [2],\*** and **Anne Dorothée Slovic [3],\***

1   Environmental Health Department, School of Public Health, University of São Paulo,
    São Paulo 01246-904, Brazil
2   Department of Epidemiology, School of Public Health, University of São Paulo, São Paulo 01246-904, Brazil
3   Environmental Health Department, Adjunct Professor and Research Fellow, School of Public Health,
    University of São Paulo, São Paulo 01246-904, Brazil
*   Correspondence: ncmaria@usp.br (N.C.M.); aralphms@usp.br (A.R.M.-S.); adslovic@usp.br (A.D.S.)

**Abstract:** The recognition of metropolitan regions and their growth as a necessary scale of analysis for their integrated management has become a central characteristic of urban planning. The current metropolitan landscape warrants the use of instruments beyond the municipal scale, especially since the economic integration of cities and their development are accentuating urban problems that affect the sustainability of cities. The São Paulo Metropolitan Region (MRSP), one of the world's megacities, is used as a case study to identify how typologies can contribute to integrated sustainable urban planning and management at the metropolitan level. It applies the territorial analytical typology based on the Driving-Force-Pressure-Situation-Exposure-Effect-Actions (DPSEEA) Environmental Health Matrix to identify the heterogeneity of conditions encountered in large metropolitan regions such as the MRSP. The results show a great variety of environmental and social conditions present in the municipalities of the MRSP that condition the sustainability and health of the urban environment. This typology constitutes a first step to characterize metropolitan regions in socioenvironmental terms using as a conceptual basis a matrix of environmental health indicators, being a precursor in the largest metropolitan region of Brazil.

**Keywords:** metropolitan region; urban planning; São Paulo; sustainability; environmental health

## 1. Introduction

Global environmental problems are related to population growth, accompanied by an accelerated world-wide process of urbanization that is not homogeneous. The environmental impacts of this process are still difficult to quantify, but the growth of the urban population creates increasing demands for infrastructure and services, resulting in an enormous lack of basic sanitation and stress over the environment and natural resources, particularly in cities from emerging economies like Brazil [1].

Studies evaluating cities that address sustainability issues have increased over the last decade as cities are now seen as an average point in the search for global sustainability. These studies serve as planning and evaluation tools for politicians, managers, and urban planners to compare different project alternatives and public policies. This helps policy-makers understand how globalization and urbanization affect urban spaces and help cities understand how they present themselves in the different dimensions of urban sustainability and identify strategic areas for improvement [2].

A sustainable system is defined as one that drives economic, environmental, and social welfare. Therefore, urban sustainability is a multidimensional concept [3]. Planning and managing growth and

urban development are key issues facing planners and policy-makers to achieve a sustainable future, especially for the world's metropolitan regions [4].

Latin America and the Caribbean regions have experienced rapid urbanization in the 20th century and currently 80% of their population is urban. The city of São Paulo, the most populated city in Brazil and the six largest in the world with its Metropolitan Region (MRSP), is considered one of the paramount urban concentrations in South America [1]. With a population of 20.9million inhabitants, the MRSP comprises 5 subregions, 39 municipalities (Figure 1), and 127 districts, out of which the municipality of São Paulo represents about 3% of the São Paulo state area and 55% of the Gross Domestic Product (GDP), higher than the whole São Paulo state [5].

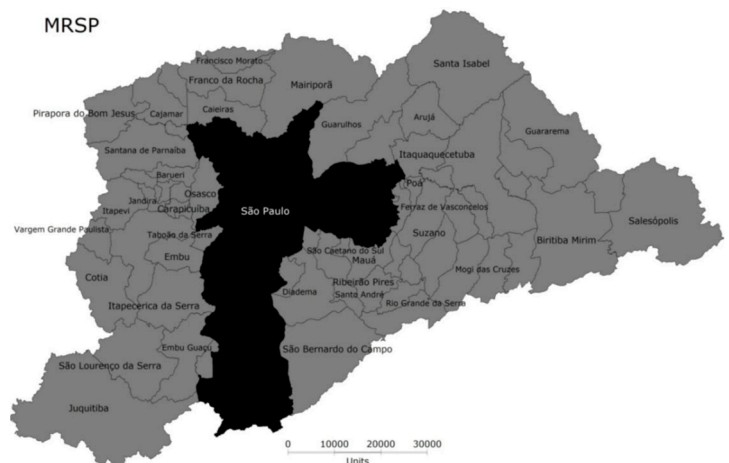

**Figure 1.** Region of São Paulo and its municipalities.

The São Paulo metropolitan region, like other major urban centers in development, has experienced a dramatic size increase since the 1950s, with an uneven spatial distribution and characterized by a periphery, in which the supply of essential public services has not kept pace with the expansion of the metropolis and the needs of its population [6].Therefore, the metropolis presents wide social inequalities and complex environmental problems reflected in degraded essential services of life support. This type of urban expansion is a feature of many Latin American cities such as Buenos Aires (AR), Santiago (CL), and particularly São Paulo [7].

The urban sprawl that involves the MRSP has significant social and economic discrepancies. Villaça (2011) [8] discusses how the process of urban segregation occurs in the production of these spatial discrepancies in both the city of São Paulo and its metropolitan region, showing the relationship between residential segregation, places of employment, the provision of transport, and more environmentally friendly areas. The author points out by indicators for these issues that 20% of the richest population of the RMSP is concentrated only in the Southwest of the São Paulo metropolis, concentrating most tertiary jobs, better transport infrastructure and green areas. Industrial jobs are concentrated in the adjacent municipalities, such as Guarulhos, Osasco, and ABC Paulista, where most of the housing of the poorest people gather in areas with less transport infrastructure and environmental conditions. Within the metropolis of São Paulo, the East Zone concentrates the majority of its poorer population. As a result of these spatial inequalities, we have 2.1 million MRSP inhabitants living in precarious housing conditions, many in risky and environmentally vulnerable areas and lacking urban and social infrastructure [9,10].These disparities appear in the other 38 metropolitan municipalities covering a region that extends over an area of 8,000 km$^2$, of which more than 50% are located in protected areas of water springs, partly occupied irregularly by clandestine settlements. The heterogeneity of socioenvironmental inequalities is a reflection of the expansion pattern of the peripheries in the process of metropolitan structuring [11]. The impacts of these inequalities, together with the municipal socioeconomic activities, created a scenario of environmental deterioration and risks to human

health [12]. As a consequence, the processes of crossed interaction and reinforcement of technological change, population growth, and urbanization have contributed to the overexploitation of ecosystems through increasingly complex feedback [13].

Technological progress reinforces population growth and urbanization; consequently, the supply of resources from local ecosystems can no longer satisfy local demand, and many of the needs that were met locally are now outsourced, resulting in an increase in the geographical extension of supply and demand. As settlements grow, the surrounding ecosystems are increasingly modified to provide services such as water, at the expense of other types of ecosystem services [13].

Thus, reconciling the demands of population growth and related settlements with sustainability is increasingly difficult. Managing sustainably requires understanding the trajectories of changes that have produced the current situation and continuing to shape them as well as the result of socioecological interactions. It encompasses the characterization of cities that make up the metropolitan region. The sustainability of the metropolitan system can be strategically used to understand the distinct aspects that involve the heterogeneity of the conditions present in regional territories [14,15].

In addition, the analysis of conditions and tendencies of the interaction of driving forces resulting from socioeconomic and environmental pressures through systems of indicators represents a challenge for Public Health [16]. Environmental issues are also health issues, given that they affect human beings and society [17]. There is growing evidence that environmental conditions play an important role in the health of the world's population. The health and well-being of urban dwellers results from complex structures that shape cities [18]. However, according to Sobral and Freitas (2010) [19], ecosystem deterioration and climate change are not decisive in urban health issues, but they adjoin some that are.

To capture the link between environmental conditions and their health effects, a variety of structures have been developed to assist in the creation and use of indicators for this area. The most cited structure in the literature for environmental health indicators is a matrix called Driving-Pressure-Situation-Exposure-Effect-Actions or DPSEEA [20], which was the methodology adopted in this article.

In order to identify the complexity of urban contexts and their environmental setting, the development of typologies appears as an important tool to classify the features that fit in a data set, thus determining types and systems that characterize the social and environmental health conditions of the municipalities; consequently, its sustainability. In addition, the development of indicators can be used as a tool to understand the metropolitan system relations, hierarchies, flows, and relations between cities.

Previous studies such as Alves et al. (2016) [21], Lopez-Carreiro and Monzon (2018) [22], Choi (2018) [23], and Jeong et al. (2016) [24] have used typologies as a tool for urban sustainable planning such as population growth, performance in intelligent transport systems, or degree of development and mobility. Holland (2015) [25] constructed a typology according to urban policies of development in North American cities, identifying eight typologies to illustrate the range of choices available to decision-makers in the policy process of urban public policy making. However, few studies to this date have used typologies in Latin American cities to explore multivariate socioenvironmental characteristics that are related to health of the environment, and none has been developed for a regional territorial cut such as the MRSP [14,15].

The purpose of this study is to propose a territorial analytical typology based on the DPSEEA Environmental Health Matrix to understand the heterogeneity of conditions that can be found in a large metropolitan region such as the MRSP, considering socioenvironmental indicators that correspond to a context of environmental sustainability in order to contribute to the perfection of an instrument to support sustainable urban planning and management.

To achieve the objectives proposed by this study, the text was divided into the following sections: Introduction presenting the area of study and the issues surrounding sustainable urban development metropolitan areas mentioning some studies in this area; Material and Methods with the selection of the indicators used and the statistical analyses employed; Presentation of Results; Discussion seeking

for an integrated approach to metropolitan area planning and management, in particular, the MRSP as a unit of analysis, and Conclusions.

## 2. Material and Methods

*Selection and Construction of Indicators*

Secondary data was obtained from documentary research in public institutions and bibliographical reviews about the theme. Indicators were selected from governmental open-access databases such as the Brazilian Institute of Geography and Statistics (IBGE), the Department of the Brazilian Unified Health System (DATASUS), and the São Paulo State Foundation of Data Analysis (SEADE).

The framework for selection and arrangement of the indicators was based on the matrix of environmental health indicators developed in partnership with the United Nations Environment Program (UNEP), the World Health Organization (WHO), and the USA Environment Program (USEPA), known as the DPSEEA framework [26]. The matrix establishes a demand flow and pressure for natural resources and changes in the ecosystem, enabling a broad observation of the factors that influence human health and well-being. Its multiple dimensions allow the analysis of the driving forces (D), the environmental pressure (P) that influence the state/situation of the environment (S), modulating the exposures (E) to diseases, which are considered the effect (EF) in the cycle. Thus, this systemic and ranked set of indicators focuses on health and environmental issues, bringing options for different and strategic actions (A) that can be executed at different levels and in different ways of prevention and control [26].

According to the environmental indicators, the manual of the Brazilian Ministry of Health [27], the model proposed by the WHO, appeared to be suitable for the rational basis exhibited in an integrated matrix of indicators conjoining socioenvironmental and health issues (Figure 2).

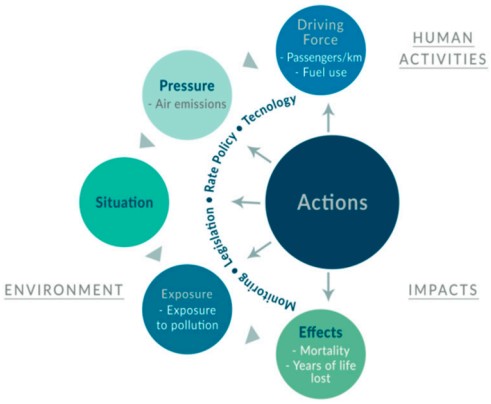

**Figure 2.** Dimensions of the DPSEEA matrix. Source: Corvalán et al. 2000 (adapted). DPSEEA: Driving-Force-Pressure-Situation-Exposure-Effect-Actions or DPSEEA.

For this project, four categories of indicators were selected: sanitation, land use, air quality, and forest coverage. These categories were chosen based on human pressure and their influence on urban environmental quality such as infrastructure, air and water quality, solid waste management, and prevalence of areas covered by natural vegetation. Each selected indicator expresses different socioenvironmental conditions in MRSP that aim at specific demands when quality and sustainability were considered.

In addition to the DPSEEA Matrix, for the selection of relevant indicators that fit the MRSP metropolitan context, the priority metropolitan themes expressed in the MRSP IUDP (Integrated Urban Development Plan) were considered. The Plan considers metropolitan issues that go beyond municipal boundaries. Examples are the use of water resources considering the possibilities of integrating the production systems and the projections of demand, availability of electricity, infrastructure barriers to

serve the population and support economic activity. With regard to sanitation and the environment, issues such as sanitary sewage, macrodrainage, solid waste disposal, and environmental damage were highlighted.

The DPSEEA approach, adopted by World Health Organization, has been applied in Brazil by the General Coordination of Environmental Health Surveillance of the Health Surveillance Secretariat of the Ministry of Health (CGVAM / SVS / MS) since the late 1990s. The construction of indicators that point to the interrelationships of environmental changes and health situation are at the origin of the process of institutionalization of environmental health surveillance within the Health Sector, from the 20th to the 21st century, initially by Decree No. 3.450/2000 of the Presidency of the Republic, establishing in the former National Center for Epidemiology (Cenepi) the management of the national system of epidemiological and environmental surveillance and, subsequently, through Normative Instruction SVS No. 1 of 2005, which regulated the National Environmental Health Surveillance Subsystem (SINVSA) [27].Brazil was a pioneer in adapting this conceptual framework and in the systematic and validated use of indicators that help manage environmental health problems [27].

Table 1 presents a set of indicators used by environmental health, organized according to the DPSEEA model:

**Table 1.** Examples of Environmental Health Indicators organized according to the DPSEEA model.

| Driving Force | Pressure | Situation | Exposure | Effect | Actions |
|---|---|---|---|---|---|
| Gini Index of the distribution of monthly income of persons 10 years of age and over with income | Sanitary sewage collection (percentage of households without sewage and/or rainwater collection service) | Inadequate sanitation (percentage of households without simultaneous water supply by mains, sewage by mains and garbage collected daily) | Tap water (percentage of people living in households without piped water for one room or more, coming from the mains, the well, the spring or the reservoir supplied by rainwater or tanker) | Infant mortality rate (number of deaths of children under one per thousand live births in the population living in geographical area in the year considered) | Existence of municipal councils of Health and Environment |
| Population growth rate (percentage difference between the population in two or more years considered) | Sewage treatment (percentage of districts without sanitary sewage treatment collected) | Garbage collection (percentage of households without regular direct or indirect garbage collection service, including burned or buried, thrown in vacant or public place, river, lake or sea and others) | Water treatment (percentage volume of water distributed per day without treatment) | Admission under 5 years by ADD1 (ADD hospitalization rate for children under 5 years and resident population of under 5 years per 1000 children) | Public expenditure on health as a proportion of GDP (percentage of gross domestic product that corresponds to public expenditure on health, broken down by government–federal, state and municipal) |
| Urbanization rate (percentage of people living in urban households) | Electricity consumption (per capita residential consumption of electricity) | Flooding or (percentage of municipalities that have experienced or flooding in the last two years | Inadequate sewage facilities (Percentage of persons living in permanent private housing units without access to adequate sewage facilities, that have a single use toilet and with sewer connected to sewage or rainfall collection system or septic tank | Admission of children under five by ARI2 for children under 5 and resident population under 5 for 1000 children) | Population coverage by PSF3 and ACS4 teams (percentage of the population residing in a given geographic space in the year which receives regular care by the PSF and ACS teams) |
| Per capita household income (total income of a family divided by number of people family) | Car fleet per inhabitant (ratio of total fleet of cars and the number of inhabitants) | Vegetation cover (percentage change of vegetation cover at different stages of regeneration) | Garbage Collection (Percentage of people living in households where garbage collection is not performed directly by a public or private company, or where garbage is not deposited in a bucket, tank, or out-of-home bin) | Mortality rate from external causes (number of deaths from external causes—accidents and violence, per 100 thousand inhabitants | - |

Source: Sobral and Freitas (2010); 1 acute diarrheal disease; 2 Acute respiratory infection; 3 Family Health Program; 4 Community Health Agents.

This model allows the integrated analysis of environmental health within an economic and social context and can be applied to subsidize the monitoring of sustainability conditions at the regional and municipal levels [19]. Stauber et al. 2018 [18] points out that applying the DPSEEA matrix structure can allow to make more focused decisions around key socioenvironmental issues by identifying areas for intervention. Thus, achieving improvements in the health and well-being of the cities requires better tracking and understanding of the role of indicators, especially in urban environments. Inserted in the logic presented, the application of the DPSEEA matrix was the basis of this study to select indicators that operationalize the concept of environmental health.

According to Sobral and Freitas (2010) [19], it is important to highlight that the DPSEEA model of indicators should be used as an auxiliary tool of the social determination model of health, because although it allows an integrated view of the indicators, by itself it cannot contemplate all the complexity of interrelations of the dimensions that determine the process of social production of health-disease and its inequalities between social groups.

Table 2 presents a detailed list of selected indicators for each theme, in accordance with the multiple dimensions of the DPSEEA matrix. For each figure, a specific definition shows the indicators and what is expressed, the institutions source of data, related system of information, and pertinent features. The process of selecting indicators was compatible with the application of the same matrix carried out by the Brazilian Ministry of Health to compare Brazilian states, adapted considering the metropolitan perimeter and the municipal scale [28].

**Table 2.** Selected indicators of the DPSEEA matrix submitted to statistical analysis.

| Dimension | Theme | Indicator | Source | Measurement Unit |
|---|---|---|---|---|
| Driving force | Soil Occupation | Population | IBGE (2010) | Number of inhabitants |
| Driving force | Soil Occupation | Population Growth Rate | SEADE (2000–2010) | Percentage |
| Driving force | Soil Occupation | Level of Urbanization | SEADE (2010) | Percentage |
| Pressure | Sanitation | People Without Sanitation (sewage or pluvial) | IBGE (2010) | Percentage |
| Pressure | Soil Occupation | Residences Subnormal Agglomeration | IBGE (2010) | Percentage |
| Pressure | Air Quality | Car Fleet Per Inhabitant | IBGE (2010) | Car Fleet, Cars/Inhabitant |
| Pressure | Vegetation Cover and Water Source | Areas of Water Stock (protected by law) | SIGAM/SMA/SP | Percentage |
| Situation | Sanitation | Residences Connected to the Public Service (sewage or pluvial) | IBGE (2010) | Percentage |
| Situation | Sanitation | Sewage Treatment Index | MINISTRY OF CITIES (2009) | Percentage |
| Situation | Sanitation | Residences With Water Distribution | IBGE (2010) | Percentage |
| Situation | Sanitation | Residences With Waste Colletion | IBGE (2010) | Percentage |
| Situation | Sanitation | Quality Index Landfill Wastes (IQR) | CETESB (2010) | |
| Situation | Sanitation | C02 Emission in Million Tones | SE/SP | Tones |
| Situation | Vegetation Cover | Native Vegetation | IF (2010) | Percentage |
| Exposure | Sanitation | Inhabitants Without Water Supply | IBGE (2010) | Nr of Houses & Residents |
| Exposure | Sanitation | People Without Sanitation | IBGE (2010) | Nr of Houses & Residents |
| Exposure | Sanitation | People Without Waste Collection | IBGE (2010) | Nr of Houses & Residents |
| Exposure | Soil Occupation | Improper Residents | IBGE (2010) | Nr of Houses & Residents |
| Effect | Sanitation | Hospitalization Due to Diarrhoea (DDA) 5 Years Old or Less | DATASUS (2006) | Nr of Hospitalization (0 to 4 years old) |
| Effect | Air Quality | Hospitalization Due to Breathing Infection (ARI) 5 Years Old or Less | DATASUS (2006) | Nr of Hospitalization (0 to 4 years old) |

It is important to emphasize that the lack of more data regarding some dimensions of the matrix such as the Effect dimension is due to the scarcity of data at the municipal level; therefore, for some

municipalities the analysis could not be performed. Most of the data was extracted from the 2010 Census, the latest Brazilian census published by the IBGE.

To set the typology, a cluster analysis in a hierarchical grouping by means of algorithm UPGMA (Unweighted Pair Group Method With Arithmetic Mean) based on Euclidian distances and an ordination analysis of Nonmetric Multidimensional Scaling (MDS) were used [29]. These analyses were performed with Paleontological Statistics software (Past Version 3.02) commonly used in biology studies [30].

The cluster analysis represents a multitude of numerical techniques whose main purpose is to classify the values of a data matrix in discreet groups. These methods are used to decide whether the objects or descriptors under study are similar enough to be allocated to a group, allowing the identification of distinctions or separations between groups [30]. One of the best-known methods of clustering is the UPGMA, which groups an object based on the average distance between this object and all members of a given cluster. The highest similarity (or smallest distance) identifies the next cluster to be formed. When two groups join, they do it based on the mean distances between all members of each group. The clustering results are generally represented as tree-like graphs called dendrograms [29,30].

Among the distance or similarity metrics used in the clustering process, one of the most common metrics is the Euclidean distance, which is the distance between two points in an $n$-dimensional space, called the Euclidean space, calculated based on the Pythagorean theorem. Thus, the following formula is used to calculate the distance between two objects described by variables:

$$d_{i,j} = \sqrt{\left(y_{i,1} - y_{j,1}\right) + \left(y_{i,2} - y_{j,2}\right) + \cdots + \left(y_{i,n} - y_{j,n}\right)} \tag{1}$$

In this formula, the compared objects are indicated by the subscript $i$ or $j$, and the variables are indicated by the numbers 1, 2, $\ldots$ , $n$.

Since the value of the Euclidean distance can be influenced by differences in the magnitude of the units of measurement of the variables, a standardization of the values is used, thus ensuring that the variables are on the same measurement scale. A commonly used standardization is the subtraction of the sample mean ($\overline{Y}$) from the value of each observation of the variable ($Y_i$), and the division of this difference by the sample standard deviation ($s$). The result of this transformation is called the Z-score:

$$Z = \frac{\left(Y_i - \overline{Y}\right)}{s} \tag{2}$$

The NMDS ordination method is used to represent ordering relationships between objects in a small and specified number of dimensions (usually two or three), without prioritizing the exact distance between objects in an ordination plot. The NMDS is a computer-intensive and iterative method that creates an ordered space where dissimilar objects are plotted far apart and similar objects are plotted close to one another [29,30].

The NMDS procedure begins with specifying the number of dimensions (or axes) sought. After this, a configuration of objects in $n$ dimensions is constructed that will be used as a starting point in an iterative adjustment process. An iterative procedure tries to position the objects in the requested number of dimensions in such a way as to minimize a stress function (scaled from 0 to 1), which measures how far the distances in the reduced-space configuration are from being monotonic to the original distance matrix of the data. The closer to 0 the value of the stress function is, the better the fit between the reduced space distance and the original data matrix distance (Euclidian distance matrix). The distances among objects in the ordination plot with the original distances can be viewed in a Shepard diagram. As with the other ordination methods, it is possible to add information coming from a clustering result to an NMDS ordination plot [29,30].

## 3. Results of Multivariate Statistical Analyses

Adendrogram was generated (Figure 3) from the cluster analysis and interpreted considering that the municipalities were grouped according to the level of derivation hierarchically in relation to the Municipality of São Paulo (MSP). In this way, types 1 and 2 are derived in relation to MSP, while type 3 is constituted by municipalities less hierarchically derived in relation to the city of São Paulo.

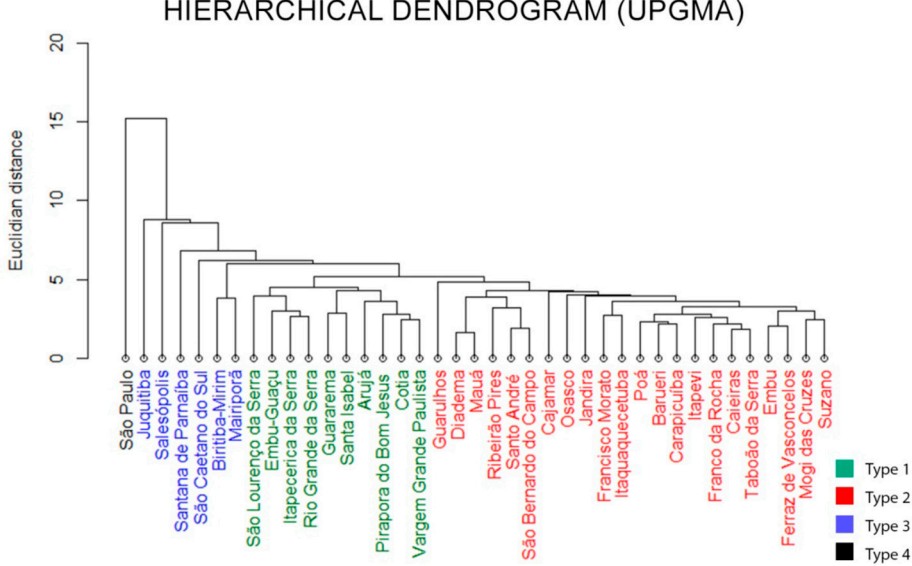

**Figure 3.** Hierarchical dendrogram (UPGMA) of the municipalities of MRSP. UPGMA: Unweighted Pair Group Method With Arithmetic Mean.

According to the cluster analysis and based on hierarchical conformation of the dendrogram (Figure 3), four types were defined and created, and observed in the NMDS scatter plot using two dimensional axes in Cartesian space (coordinates 1 and 2) (Figure 4).

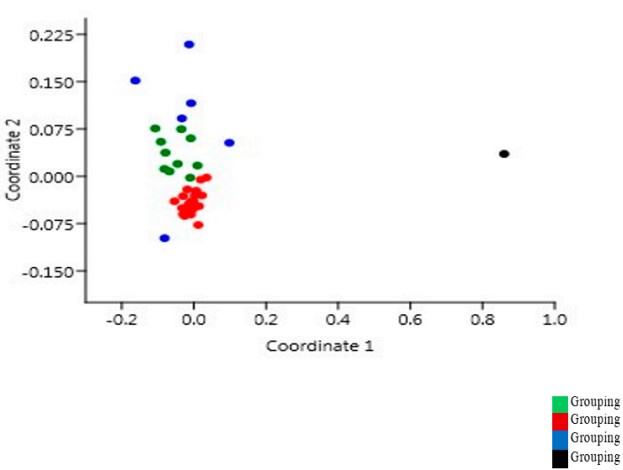

**Figure 4.** Nonmetric Multidimensional scaling (MDS) of the four types defined for the municipalities of the MRSP plotted in the space of the main coordinates 1 and 2.

The stress function (scaled from 0 to 1) was calculated to obtain the value of 0.2114. This value showed how far the distances in the reduced-space configuration are from being monotonic to the original distance matrix of the data.

The distances between objects in the ordination plot with the original distances are presented in the Shepard diagram below (Figure 5):

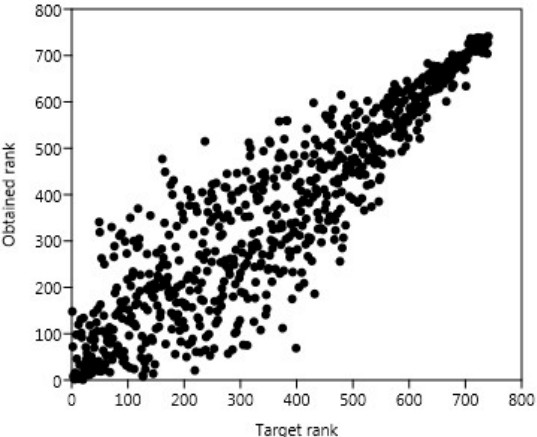

**Figure 5.** Shepard diagram.

The NMDS analysis showed that all the points were separated from the point representing the city of São Paulo (black). The points in groupings 1 (green) and 2 (red) are displayed closer in the coordinate space. Grouping 3 (blue) presents its scattered points between groupings 1 and 2. Therefore, grouping 3 consists of several groups external to groupings 1 and 2, due to the fact that a hierarchical approach for the dendrogram was used.

Besides the many subdivisions in smaller groups formed in each main cluster of the dendrogram, it is possible to observe that there are more subdivisions among municipalities as their heterogeneity increases. The four groupings of municipalities are also visibly gathered in a thematic map (Figure 6) and can be comparatively analyzed to discuss the selected indicators.

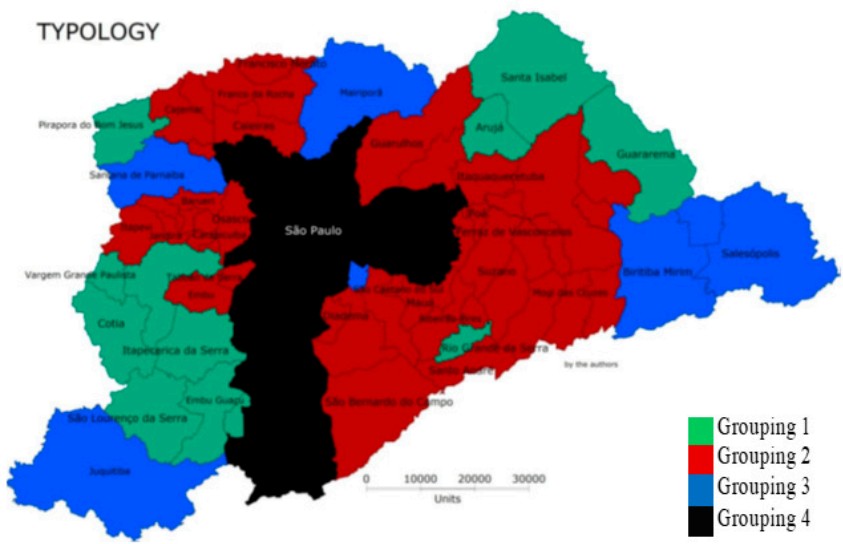

**Figure 6.** Thematic map representing the four groupings of municipalities.

## 3.1. Socioenvironmental Groupings of Municipalities in the MRSP

### 3.1.1. Grouping 1—Environmental Service Providers with Low Infrastructure Coverage

The firstgrouping (Table 3) includes themunicipalitiesof: Arujá, Cotia, Vargem Grande Paulista, Pirapora do Bom Jesus, Guararema, Santa Isabel, Embu-Guaçu, Itapecerica da Serra, Rio Grande da Serra, and São Lourenço da Serra. The grouping included lower-rate households connected to the water supply system and to sewage collection. However, these communities can be considered to be environmental suppliers due to the sizable area of forest coverage. In this grouping, all the municipalities showed a percentage of native vegetation, and seven of them had a high percentage of areas of water stock protected by law.

**Table 3.** Indicators indexed by quartiles and organized according to the sequence of subgroups resulting from the analysis (UPGMA).

High
Low

| Grouping 1 | | | | | | | | | | |
|---|---|---|---|---|---|---|---|---|---|---|
| Municipalities | DF.1 Population (hab 2010) | DF.2 Pop. Growth rate | DF. 8 Level of urbanization (2010%) | P5. Residences subnormal agglomeration (%) 2010 | EX.5 Improper residents (2010) | P2. Car fleet per inhabitant (2010) | S9. $CO_2$ emission in million tones 2010 | EF.2 Hospitalization due to breathing infection (2006) | Areas of Water Stock (protected by law) | Native vegetation percentage (2010) |
| Arujá | 74905 | 2.41 | 96.01 | 0 | 54 | 0.5 | 0.0924 | 3.7 | 51 | 25.3 |
| Cotia | 201150 | 3.05 | 100 | 0.6 | 292 | 0.43 | 0.2029 | 15.1 | 65 | 45 |
| Vargem Grande Paulista | 42997 | 2.83 | 100 | 0 | 40 | 0.38 | 0.0425 | 14 | 0 | 23.7 |
| Pirapora do Bom Jesus | 15733 | 2.44 | 100 | 0 | 0 | 0.28 | 0.0056 | 17.1 | 0 | 29.2 |
| Guararema | 25844 | 1.67 | 86.05 | 0 | 131 | 0.38 | 0.0425 | 12.2 | 0 | 14.8 |
| Santa Isabel | 50453 | 1.44 | 78.47 | 0 | 113 | 0.34 | 0.0533 | 10.8 | 82 | 23.2 |
| Embu-Guaçu | 62769 | 1.02 | 97.33 | 0 | 861 | 0.31 | 0.0434 | 13.7 | 100 | 41.6 |
| Itapecerica da Serra | 152614 | 1.67 | 99.17 | 0.91 | 144 | 0.3 | 0.2073 | 8.1 | 100 | 44.5 |
| Rio Grande da Serra | 43974 | 1.72 | 100 | 0 | 322 | 0.25 | 0.0054 | 10 | 100 | 56.3 |
| São Lourenço da Serra | 13973 | 1.4 | 91.02 | 0 | 20 | 0.33 | 0.0335 | 6.7 | 100 | 66.3 |

| Grouping 1 | | | | | | | | | | |
|---|---|---|---|---|---|---|---|---|---|---|
| Municipalities | P.1 People without sanitation (sewage or pluvial) (%) 2010 | S.1Residents connected to the public service (sewage or pluvial) (%) 2010 | S.2 Sewage treatment index (%) 2009 | S.6 Residences with water distribution (%) 2010 | S.3 Residences with waste collection (%) 2010 | S.4 Quality index landfill wastes 2010 | EX.1 Inhabitants without water supply (2010) | EX.2 Inhabitants without sanitation (2010) | EX.3 Inhabitants without waste collection (2010) | EF.1 Hospitalization due to diarrhea (DDA) 5 years old or less (2006) |
| Arujá | 44 | 56.28 | 97 | 93 | 99.3 | 8.2 | 2986 | 9619 | 285 | 0.5 |
| Cotia | 47 | 52.94 | 27.3 | 90.98 | 99.29 | 9.4 | 9446 | 29998 | 875 | 1.8 |
| Vargem Grande Paulista | 53 | 46.8 | 0 | 92.45 | 99.47 | 9.4 | 5398 | 6881 | 261 | 1.1 |
| Pirapora do Bom Jesus | 36 | 63.82 | 54 | 92.22 | 97.06 | 8.9 | 8 | 212 | 2 | 1 |
| Guararema | 56 | 44.1 | 31.4 | 68.64 | 97.73 | 9 | 4188 | 5546 | 232 | 4.3 |
| Santa Isabel | 45 | 54.92 | 0 | 71.99 | 96.08 | 9 | 3878 | 5158 | 195 | 2.2 |
| Embu-Guaçu | 59 | 40.74 | 100 | 77.32 | 97.87 | 8.2 | 12725 | 25960 | 1140 | 3.5 |
| Itapecerica da Serra | 65 | 35.5 | 57 | 89.68 | 98.86 | 8.2 | 3709 | 60037 | 710 | 1 |
| Rio Grande da Serra | 39 | 61.16 | 85 | 95.25 | 98.69 | 9.4 | 2134 | 9913 | 563 | 1.3 |
| São Lourenço da Serra | 59 | 40.97 | 100 | 57.64 | 94.55 | 8.2 | 120 | 356 | 27 | 0.6 |

3.1.2. Grouping 2—Better Infrastructure Coverage with Greater Social Inequalities

The secondgrouping (Table 4) isthe largestoneandencompasses 22 municipalities: Barueri, Carapicuíba, Poá, Caieiras, Taboão da Serra, Franco da Rocha, Itapevi, Embu, Ferraz de Vasconcelos, Mogi das Cruzes, Suzano, Francisco Morato, Itaquaquecetuba, Jandira, Osasco, Cajamar, Diadema, Mauá, Ribeirão Pires, Santo André, São Bernardo do Campo, and Guarulhos. Grouping 2 showed the best indicator for households connected to the sewage system, with a more consistent urban infrastructure. However, even with good indicators for infrastructure, this grouping presented high values for indicators of social inequities. The following municipalities stood out: Itapevi, Mogi das Cruzes, Suzano, Francisco Morato, Itaquaquecetuba, Osasco, and Guarulhos. According to the forest coverage indicators, 12 of these municipalities had a high value of native vegetation, but the values of territorial percentage of municipal area were smaller than in the first type. Suzano, Jandira, and Ribeirão Pires showed a greater value for areas of water stock protected by law.

**Table 4.** Indexed by quartiles and organized according to the sequence of subgroups resulting from the analysis (UPGMA).

High

Low

| Grouping 2 | | | | | | | | | | |
|---|---|---|---|---|---|---|---|---|---|---|
| Municipalities | DF.1 Population (hab 2010) | DF.2 Pop. Growth rate | DF. 8 Level of urbanization (2010%) | P5. Residences subnormal agglomeration (%) 2010 | EX.5 Improper residents (2010) | P2. Car fleet per inhabitant (2010) | S9. $CO_2$ emission in million tones 2010 | EF.2 Hospitalization due to breathing infection (2006) | Areas of Water Stock (protected by law) | Native vegetation percentage (2010) |
| Barueri | 240749 | 1.49 | 100 | 0.93 | 70 | 0.49 | 0.4369 | 16.2 | 0 | 8.8 |
| Carapicuíba | 369584 | 0.72 | 100 | 7.11 | 78 | 0.34 | 0.4010 | 17.1 | 0 | 3.2 |
| Poá | 106013 | 1.03 | 98.42 | 0 | 47 | 0.31 | 0.1089 | 11.8 | 6 | 5.4 |
| Caieiras | 86529 | 2.01 | 97.52 | 2.64 | 36 | 0.3 | 0.2127 | 9.9 | 20 | 16.4 |
| Taboão da Serra | 244528 | 2.15 | 100 | 10.16 | 9 | 0.33 | 0.1819 | 15.6 | 0 | 9.2 |
| Franco da Rocha | 131604 | 1.99 | 92.13 | 6.95 | 164 | 0.23 | 0.0590 | 12.7 | 5 | 14 |
| Itapevi | 200769 | 2.16 | 100 | 1.48 | 499 | 0.27 | 0.0948 | 15.3 | 0 | 22.6 |
| Embu | 240230 | 1.48 | 100 | 13.14 | 165 | 0.27 | 0.2890 | 14.5 | 59 | 23.8 |
| Ferraz de Vasconcelos | 168306 | 1.71 | 95.51 | 6.41 | 133 | 0.22 | 0.0632 | 20.3 | 40 | 24.5 |
| Mogi das Cruzes | 387779 | 1.62 | 92.14 | 0 | 836 | 0.43 | 0.4868 | 20.2 | 49 | 22.8 |
| Suzano | 262480 | 1.41 | 96.48 | 1.92 | 851 | 0.31 | 0.5773 | 12.1 | 72 | 17.2 |
| Francisco Morato | 154472 | 1.48 | 99.8 | 5.38 | 435 | 0.15 | 0.0481 | 17.6 | 0 | 16.7 |
| Itaquaquecetuba | 321770 | 1.69 | 100 | 8.26 | 1318 | 0.18 | 0.2022 | 9 | 0 | 9.5 |
| Jandira | 108344 | 1.69 | 100 | 1.78 | 1 | 0.34 | 0.0902 | 19.7 | 100 | 6 |
| Osasco | 666740 | 0.23 | 100 | 10.65 | 894 | 0.45 | 0.8286 | 12.8 | 0 | 2.3 |
| Cajamar | 64114 | 2.38 | 97.99 | 4.13 | 92 | 0.34 | 0.2127 | 35.8 | 0 | 12.8 |
| Diadema | 386089 | 0.79 | 100 | 20.97 | 121 | 0.36 | 0.3399 | 24.1 | 22 | 4.8 |
| Mauá | 417064 | 1.4 | 100 | 18.25 | 139 | 0.37 | 0.4567 | 16.2 | 19 | 10.9 |
| Ribeirão Pires | 113068 | 0.8 | 100 | 2.63 | 213 | 0.47 | 0.1012 | 13.2 | 100 | 30.7 |
| Santo André | 676407 | 0.41 | 100 | 11.04 | 9 | 0.62 | 1.1258 | 10.8 | 54 | 35.8 |
| São Bernardo do Campo | 765463 | 0.87 | 98.33 | 18 | 124 | 0.58 | 1.1372 | 16 | 53 | 47 |
| Guarulhos | 1221979 | 1.33 | 100 | 15.98 | 714 | 0.36 | 6.2046 | 11.5 | 0 | 29.5 |

**Table 4.** *Cont.*

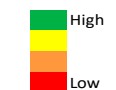

High
Low

| | | | | | Grouping 2 | | | | | |
|---|---|---|---|---|---|---|---|---|---|---|
| Municipalities | P.1 People without sanitation (sewage or pluvial) (%) 2010 | S.1 Residents connected to the public service (sewage or pluvial) (%) 2010 | S.2 Sewage treatment index (%) 2009 | S.6 Residences with water distribution (%) 2010 | S.3 Residences with waste collection (%) 2010 | S.4 Quality index landfill wastes 2010 | EX.1 Inhabitants without water supply (2010) | EX.2 Inhabitants without sanitation (2010) | EX.3 Inhabitants without waste collection (2010) | EF.1 Hospitalization due to diarrhea (DDA) 5 years old or less (2006) |
| Barueri | 9 | 90.95 | 4.5 | 99.26 | 99.92 | 8.9 | 1431 | 10202 | 226 | 3.3 |
| Carapicuíba | 19 | 81.19 | 16.2 | 98.1 | 99.48 | 8.9 | 5398 | 36976 | 1122 | 3.2 |
| Poá | 6 | 93.63 | 53 | 98.9 | 99.58 | 8.2 | 458 | 3359 | 220 | 3.1 |
| Caieiras | 15 | 85.47 | 0 | 96.41 | 99.76 | 8.2 | 2043 | 6179 | 112 | 2.6 |
| Taboão da Serra | 9 | 90.82 | 11.7 | 99.63 | 99.94 | 8.2 | 939 | 15055 | 130 | 2 |
| Franco da Rocha | 32 | 67.68 | 0 | 94.87 | 98.22 | 8.2 | 1367 | 23070 | 732 | 1.2 |
| Itapevi | 26 | 74.44 | 0 | 95.06 | 99.22 | 9.4 | 10197 | 33130 | 1664 | 2.7 |
| Embu | 27 | 72.78 | 55 | 98.01 | 99.32 | 7.6 | 4178 | 41122 | 1383 | 1.8 |
| Ferraz de Vasconcelos | 17 | 83.25 | 56 | 98.37 | 99.12 | 8.2 | 1418 | 20243 | 919 | 3.3 |
| Mogi das Cruzes | 23 | 76.98 | 42.1 | 89.74 | 98.26 | 8.2 | 16090 | 38469 | 1634 | 4.4 |
| Suzano | 18 | 82.24 | 70 | 92.29 | 98.8 | 8.2 | 13396 | 23877 | 1934 | 0.5 |
| Francisco Morato | 50 | 49.91 | 0 | 95.78 | 97.89 | 8.2 | 5244 | 56534 | 2541 | 2.3 |
| Itaquaquecetuba | 29 | 71.33 | 7 | 96.69 | 98.81 | 8.2 | 10731 | 59354 | 3873 | 1.4 |
| Jandira | 17 | 83.5 | 0 | 99.37 | 99.93 | 9.4 | 721 | 10629 | 93 | 7 |
| Osasco | 16 | 83.76 | 20.5 | 99.12 | 99.29 | 7 | 2784 | 49138 | 3049 | 2.3 |
| Cajamar | 30 | 70.32 | 0 | 90.78 | 99.01 | 8.2 | 3331 | 11249 | 211 | 4.7 |
| Diadema | 3 | 96.55 | 12.8 | 99.43 | 99.61 | 9.4 | 2277 | 8323 | 1171 | 3 |
| Mauá | 10 | 90.36 | 3.7 | 99.31 | 99.8 | 9.4 | 2289 | 27741 | 764 | 2.9 |
| Ribeirão Pires | 19 | 80.7 | 70 | 95.17 | 99.52 | 9.4 | 4777 | 13189 | 386 | 1.5 |
| Santo André | 6 | 94.48 | 26.1 | 97.61 | 99.91 | 9.4 | 12979 | 19701 | 620 | 1.9 |
| São Bernardo do Campo | 11 | 89.09 | 25.6 | 98.06 | 99.83 | 9.4 | 4070 | 33820 | 730 | 1.8 |
| Guarulhos | 13 | 86.9 | 0 | 97.6 | 99.65 | 9.8 | 21455 | 54745 | 1842 | 1.6 |

### 3.1.3. Grouping 3—Environmental Service Providers with Low Infrastructure Coverage

The third grouping (Table 5) gathered the following municipalities: Biritiba-Mirim, Mairiporã, São Caetano do Sul, Santana de Parnaíba, Salesópolis, and Juquitiba, which presented diverse hierarchical levels of proximity to outermost group (São Paulo municipality). Each of these municipalities showed specific features. Only Biritiba-Mirim and Mairiporã are similar. Municipalities of this grouping, as well as the ones of Grouping 1, can be characterized by infrastructure indicators with lower percentage of households connected to a sewage system, water supply system and garbage collection. The only exception is São Caetano, that held the best values for all the selected indicators in the metropolitan area. A particularity of Grouping 3 is the higher-value Acute Respiratory Infection (ARI) incidence indicator. Another major aspect is thatBiritiba-Mirim, Mairiporã, Salesópolis, and Juquitiba can be considered important environmental suppliers, with a high percentage of native vegetation and areas of water stock protected by law. These municipalities house important river sources that supply both the city of São Paulo and its metropolitan area.

**Table 5.** Indicators indexed by quartiles and organized according to the sequence of subgroups resulting from the analysis (UPGMA).

High

Low

### Grouping 3

| Municipalities | DF.1 Population (hab 2010) | DF.2 Pop. Growth rate | DF. 8 Level of urbanization (2010%) | P5. Residences subnormal agglomeration (%) 2010 | EX.5 Improper residents (2010) | P2. Car fleet per inhabitant (2010) | S9. $CO_2$ emission in million tones2010 | EF.2 Hospitalization due to breathing infection (2006) | Areas of Water Stock (protected by law) | Native vegetation percentage (2010) |
|---|---|---|---|---|---|---|---|---|---|---|
| Biritiba-Mirim | 8575 | 1.51 | 85.83 | 0 | 36 | 0.24 | 0.0126 | 29.7 | 89 | 27.1 |
| Mairiporã | 80956 | 3.04 | 87.39 | 0 | 747 | 0.39 | 0.0692 | 34.6 | 80 | 39.5 |
| São Caetano do Sul | 149263 | 0.62 | 100 | 0 | 0 | 0.79 | 0.2256 | 26.1 | 0 | 0 |
| Santana de Parnaíba | 108813 | 3.85 | 100 | 3.48 | 33 | 0.48 | 0.0879 | 10.8 | 0 | 24.2 |
| Salesópolis | 15635 | 0.87 | 63.66 | 0 | 0 | 0.43 | 0.0128 | 23 | 0 | 34.6 |
| Juquitiba | 8737 | 0.85 | 77.39 | 0 | 103 | 0.28 | 0.0987 | 10.3 | 98 | 72.6 |

### Grouping 3

| Municipalities | P.1 People without sanitation (sewage or pluvial) (%) 2010 | S.1 Residents connected to the public service (sewage or pluvial) (%) 2010 | S.2 Sewage treatment index (%) 2009 | S.6 Residences with water distribution (%) 2010 | S.3 Residenceswith waste collection (%) 2010 | S.4 Quality index landfill wastes 2010 | EX.1 Inhabitants without water supply (2010) | EX.2 Inhabitants without sanitation (2010) | EX.3 Inhabitants without waste collection (2010) | EF.1 Hospitalization due to diarrhea (DDA) 5 years old or less (2006) |
|---|---|---|---|---|---|---|---|---|---|---|
| Biritiba-Mirim | 40 | 60.36 | 61 | 65.57 | 94.49 | 8.2 | 2171 | 1481 | 136 | 3.2 |
| Mairiporã | 65 | 35.41 | 62 | 70.71 | 95.57 | 8.2 | 16434 | 18788 | 1063 | 7.1 |
| São Caetano do Sul | 0 | 99.85 | 80 | 99.96 | 100 | 9.4 | 50 | 11 | 5 | 4.7 |
| Santana de Parnaíba | 57 | 42.6 | 2.6 | 90.88 | 99.47 | 4.9 | 2361 | 16805 | 95 | 0.9 |
| Salesópolis | 46 | 54.49 | 90 | 62.86 | 96.58 | 8.2 | 560 | 593 | 4 | 18.1 |
| Juquitiba | 78 | 22.12 | 100 | 58.59 | 86.65 | 8.2 | 820 | 2502 | 231 | 0.6 |

### 3.1.4. Grouping 4—São Paulo (MSP)

The MSP (Table 6) is considered a grouping in itself. For water supply and sanitation indicators, São Paulo and São Caetano do Sul were similar with the best values in the whole metropolitan area. However, the municipality has the largest population (12 million inhabitants) and the greatest social disparities, characterized by a deficit in public services distributed to the bottom of its districts. This can be observed in the high values of the following indicators: residents with inadequate housing and higher number of inhabitants with no water supply. It also has the highest indicator of $CO_2$ emissions, the highest in the entire metropolitan area. Two large reservoirs serve as water supply to the São Paulo municipality: Billings and Guarapiranga. Belonging to the Alto Tietê river basin, Billings and Guarapiranga are both affected by environmental degradation due to illegal settlements. With the growth of the metropolis and water demand and degradation, both reservoirs have become important environmental areas forcing the MSP to import large amounts of water to meet its population demands. Due to the urban sprawl within the watershed-protected areas, the illegal settlements have been significant contributors to this degradation. Regarding ecosystems, 21.3% of São Paulo's territory is covered with remnants of native rain forests and 36% are watershed-protected areas.

**Table 6.** Indicators indexed by quartiles and organized according to the sequence of subgroups resulting from the analysis (UPGMA).

High
(yellow)
(orange)
Low

| Grouping 4 | | | | | | | | | |
|---|---|---|---|---|---|---|---|---|---|
| **Municipalities** | DF.1 Population (hab 2010) | DF.2 Pop. Growth rate | DF. 8 Level of urbanization (2010%) | P5. Residences subnormal agglomeration (%) 2010 | EX.5 Improper residents (2010) | P2. Car fleet per inhabitant (2010) | S9. $CO_2$ emission in million tones 2010 | EF.2 Hospitalization due to breathing infection (2006) | Areas of Water Stock (protected by law) | Native vegetation percentage (2010) |
| São Paulo | 11253503 | 0.76 | 99.1 | 9.95 | 3258 | 0.59 | 12.7739 | 12,7739 21,80 | 36 | 21.3 |

| Grouping 4 | | | | | | | | | |
|---|---|---|---|---|---|---|---|---|---|
| **Municipalities** | P.1 People without sanitation (sewage or pluvial) (%) 2010 | S.1 Residents connected to the public service (sewage or pluvial) (%) 2010 | S.2 Sewage treatment index (%) 2009 | S.6 Residences with water distribution (%) 2010 | S.3 Residences with waste collection (%) 2010 | S.4 Quality index landfill wastes 2010 | EX.1 Inhabitants without water supply (2010) | EX.2 Inhabitants without sanitation (2010) | EX.3 Inhabitants without waste collection (2010) | EF.1 Hospitalization due to diarrhea (DDA) 5 years old or less (2006) |
| São Paulo | 8 | 91.86 | 81.1 | 99.09 | 99.79 | 9,3 e 8,2 | 72929 | 581236 | 15334 | 3.3 |

## 4. Discussion

### 4.1. The Search for Indicators and Typologies to Characterize Metropolitan Areas

This research has demonstrated the complexity and challenges that emerge in addressing the MRSP through a system of established indicators. Which indicators are representative of the quality of the environment in urban areas? Are they related to sustainability? These responses are not consensual and depending on the methodological approach, different results are attained.

The quest for sustainability leads to strategic formulations that are aimed to result in public policy. These, in turn, must have measurement parameters so that the monitoring, overtime, is carried out, partly due to the use of indicators. Furthermore, defining a condition of sustainable development consists in operationalization of a concept and indicators [31]. Regarding environmental issues, for Bellen (2006) [32], the lack of consensus of a unified definition within this field of study creates a challenge when identifying indicators. The vast majority of existing and used indicator systems have been developed for specific reasons and cannot be considered indicators of sustainability per se due to their lack of interconnection. However, they often have representative potential within the context of sustainable development.

The set of indicators selected referred to the environmental quality of municipalities, reflecting their environmental health characteristics and their sustainability. Environmental quality contributes significantly to social welfare, public health, and urban sustainability. In this context, urban sustainability can be defined as a dimension of development, since it represents the possibility of ensuring sociopolitical changes that do not compromise the ecological and social systems in which communities are sustained [33].

In addition to the social and economic indicators used in the analysis of health situations, the DPSEEA model allows the incorporation of environmental indicators related to the ecological integrity of the ecosystems, acknowledging the importance that environmental quality and services provided by ecosystems have to achieve health and well-being [19].

The Millennium Ecosystem Assessment (2005) [34] defines services as the benefits that people get from the ecosystem. It is possible to include provision services such as food and water; service regulations such as preventing floods, severe droughts, soil deterioration and diseases; supporting services such as soil formation and nutrient cycling; cultural services such as leisure, spiritual, religious and other nonmaterial benefits. Therefore, the extensive vegetative coverage, especially the native forest located in regions of water sources or watersheds, is a provider of ecosystem services. In this study, to identify municipalities that provide ecosystem services, indicators of the percentage of areas of protected water sources and native vegetation were used. However, no indicator was used to identify ecosystem degradation. For indicators of carbon emissions that contribute to unsustainability, the indicator of $CO_2$ emissions in millions of tons was provided by the Secretary of Energy of the State of São Paulo.

### 4.2. Limitations

The use of indicators as support for the development of public policies carries some challenges and limitations that emerged throughout the development of this work. For instance, economic indicators such as GDP were not included in the statistical analysis given that they are not reliable for evaluating other dimensions that are crucial for social and environmental development [35,36].

Regarding economic indicators, the 1980s were the most widespread socially. Well-known indicators such as Gross Domestic Product (GDP), interest rate, public deficit, and other macroeconomic statistics are widely used in decision-making in public and private instances. However, economic indicators do not respond to the need to measure and therefore assess other crucial dimensions of development, such as social and environmental, and may only represent economic growth [25].

Veiga (2010) [36], citing the "Report by the Commission on the Measurement of Economic Performance and Social Progress"—by the Stiglitz–Sen–Fitoussi Commission (2009), notes that it is one

thing to measure performance, another to measure quality of life (welfare), and a third is to measure sustainability. For the author, it is important to keep in mind the recommendations of the document on sustainability that assume that economic development and quality of life are measured by new indicators that have nothing to do with current GDP and HDI (Human Development Index). According to Martínez-Alier (2007) [37], there is a great complexity when trying to reconcile economic expansion and conservation of the environment, as social and environmental problems arise as considerations of suggestion or third order when, in modern societies, the growth of Gross Domestic Product continues being the most used reference.

Finding values that encompass the environmental aspects of sustainability requires the search for nonmonetary indicators that approach the dangerous levels of environmental damage, such as those associated with climate change. According to Veiga (2010) [36], well-calculated indicators for carbon emissions could be indicators of contributions to unsustainability, as well as similar measures for water resource impairment and biodiversity erosion. For the author, perhaps, these sets of indicators would suffice to show how far off the path of sustainability is.There is a difficulty in finding environmental indicators related to ecosystems and natural resources according to Freitas et al. (2007) [38]. That is due to the scarcity of indicators for this area and the fragmented view of the relationship between health and environment expressed by the feeble scientific production that considers the interface between ecosystems and human health.

Regarding health indicators, the most traditional ones were selected, such as sanitation (water supply and sanitary sewage collection and garbage collection), and morbidity as hospitalization rates for infectious and respiratory diseases. As observed, many environmental indicators are still restricted to sanitation, which made it difficult to analyze the environmental situation of municipalities in the metropolitan area. Another challenge was the difficulty in finding indicators related to air quality, one of the world's greatest environmental health challenges. The Environmental Company of the State of São Paulo's (CETESB) monitoring network focuses on the center of the city of São Paulo but lacks air quality data for each municipality, limiting to general values for the MRSP as a whole.

Considering the period studied, the selected indicators mirrored the interval between the years 2005 and 2010 since most of the data adopted comes from the 2010 Census. Moreover, many indicators could not be considered because they did not have values for all. In this study, the DPSEEA matrix could be applied as an auxiliary tool for selecting environmental health indicators with a potential in the urban sustainability context to compose the typology. However, the interrelations of the matrix dimensions that determine and mediate the process of social production of health-disease could not be interpreted, since there are gaps in indicators, which prevented them from being considered in the analysis, making it impossible for the dimensions to be expressed in totality. Thus, as more and better information becomes available in public systems, these methodological gaps are expected to lower in the future.

*4.3. Building a New Approach*

Contrary to the frequent precarious overlap in socioeconomic, housing and public service quality, environmental conditions do not necessarily move in the same direction. There is spatial coincidence of urban precariousness and remnants of green areas, where the main areas providing ecosystem services are located. This could be observed in the case of municipalities of types 1 and 3 with indicators of smaller values for sanitation, but with the highest values for the indicators of water stock and native vegetation protected areas. On the other hand, type 2 municipalities have better values for sanitation indicators, but they reveal a larger social inequality and, in relation to the indicators of vegetation cover, have lower values.

The two indicators that represent the provision of ecosystem services in established clusters, percentage of stock of water areas protected by laws, and percentage of native vegetation are areas established by the public power with specific legislation.

The percentage of native vegetation cut in each municipality provided by the Forest Institute of the State of São Paulo (FI) provides a survey of the Integral Protection Units and Units of Sustainable Use that shelters categories as to the form of protection and permitted uses regulated by the National System of Conservation Units (NSCU) Law Nº. 9.985/2000 (National System of Conservation Units, 2011) [39].

The fragmentation of the metropolis into administrative units based on political and economic principles differs from the spatial distribution of ecosystem characteristics. According to Steiner (2004) [40], city limits can be natural, political, and administrative, thus political borders may or may not correspond to natural borders. This lack of correspondence between natural and administrative systems, however, entails problems of resilience of urban environments to ecological phenomena.

There is an environmental interconnectivity among the municipalities since the forest remnants and the water source areas are connected across borders. This may reverse the hierarchy of an economic centrality when we try to prioritize public policies for environmental preservation. We know that there is a flow of economic and social exchanges between the municipality of São Paulo—the main economic pole of the metropolitan system—and the other cities that constitute it. However, cities that have few environmental resources in their territories have an interdependence of environmental resources from other localities.

Interconnected municipal forest fragments allow gene flow between fauna and flora species, enhancing biodiversity conservation. In addition, they guarantee water resources and soil conservation, including the balance of climate and landscape. These remnants maintain the environmental balance of the metropolitan system and the survival of large cities that consume many resources to keep the economic system in operation. The applied analysis of a metropolitan region can contribute to expand the understandings over megacities, those that can be interpreted as socioecological systems in which there is a need to apprehend cross-scale interactions and a sort of dynamics and centralities characterizing a panarchy (Walker et al. 2004) [41], in the sense of multiple domains of providing necessary ecosystem services as well as demands for natural resources.

### 4.4. Integrated Approach

In establishing the typology, we were able to obtain a general view of the MRSP by observing through the selected indicators the distribution of some of its social and environmental characteristics. Areas with better infrastructure and areas with urban precariousness in socioeconomic terms could be observed, but in the meantime, they have environmental resources. As there is an environmental interconnectivity among municipalities, identifying these characteristics is of strategic importance for metropolitan sustainability.

In Brazil, policies for public management are local and centralized with municipal autonomy, since the 1988 Federal Constitution decentralized municipal resources and competences prioritizing local autonomy. However, this political centralization can affect the sustainability of regional systems, as municipalities become increasingly interdependent. Using the regional scale as a planning and management unit offers the possibility of gathering answers beyond the municipal scale, especially because of the problems caused by economic integration of cities, such as environmental degradation, lack of basic sanitation, unemployment, lack of urban infrastructure, and violence arising from socio-spatial segregation have become more intense. The regional scale is one way of saying that the problems in restructuring these areas affect more than one city, thus fomenting the political discussion about municipal spaces integrated and marked by common institutional challenges.

Examples of environmental problems of metropolitan magnitude include the case of air pollution not confined to a municipality and water scarcity, since municipalities with larger water sources, which offer important ecosystem services, are mostly located in the periphery of the São Paulo Metropolitan Region. Disregarding these issues could lead to impairment of water production.

The discussion about the legalization of instruments that guide the planning and management of metropolitan areas is recent in Brazil, as can be seen from the publications of current legislation.

The Metropole Statue, Federal Law Nº. 13.089 [42] establishes the legislation and general guidelines for the planning, management, and execution of public functions of common interest in metropolitan regions and state-established urban agglomerations, and general rules on integrated urban development and other instruments of intercorporate governance. The Metropole Statute was sanctioned in 2015 and modified by a provisional measure Nº. 818 in 2018.

The Statute established an integrated development of metropolitan regions and urban agglomerations via the following instruments:

I.　　　Integrated Urban Development Plan (IUDP);

II.　　　Interfederative Sector Plans;

III.　　　Public Funds;

IV.　　　Interfederated Urban Consortium Operations;

V.　　　Zones for shared application of urban planning instruments provided for in Law Nº. 10.257, of 10 July 2001;

VI.　　　Public Consortia, pursuant to Law Nº. 11.107, of 6 April 2005;

VII.　　　Cooperation Agreements;

VIII.　　　Management Contracts;

IX.　　　Compensation for environmental services or other services rendered by the municipality to the urban territorial unit;

X.　　　Interfederative Public–Private Partnerships.

Of all the mentioned instruments, the main one to promote integrated urban development are the IUDPs because they are responsible for making the City Master Plans compatible with the integrated urban development of the urban territorial unit to which they belong. The legislation required that all metropolitan areas and Brazilian urban agglomerations develop, until 31 December 2021, their Urban Development Integrated Plans (IUDPs). After its approval, the municipalities that integrate these territorial units must reconcile their Municipal Director plans with the new rules (Urban Development Integrated Plan of MRSP, 2016) [43].

The MRSP IUDP establishes the guidelines to urban and regional development of the territory following three developmental axes: territorial and inclusive urban cohesion; territorial connectivity and economic competitiveness and metropolitan governance.

Among the strategic guidelines proposed by the IUDP is the structuring of a network of metropolitan poles to improve the quality of life in areas more distant from the more consolidated urban centers. With respect to the physical-territorial dimension closely related to the sustainability of the ecosystem services of the regional territories, the IUDP mentions that the orientation of the urban occupation, the intensification of the use of the idle urban areas, the improvement of the distribution of the activities in the territory, the guarantee of supply of water for future generations, the promotion of ecological corridors to maintain biodiversity and preserve water sources, and the guarantee of a collectively constructed legal framework are guidelines of the plan as a whole. This demonstrates how important the environmental dimension is in issues involving the planning and management of regional territories to ensure the sustainability of natural resources and the quality of life of populations (Urban Development Integrated Plan of MRSP, 2016) [43].

The typology in Environmental Health is in line with the integrated development instruments established by metropolitan legislation as it considers indicators related to two important issues of the IUDPs: the preservation of the environmental heritage and the ability to produce ecosystem services alongside with the reduction of inequalities. The typology allows a visualization of the plurality of characteristics of a heterogeneous development that can be found within the municipalities that compose the MRSP.

Using as a territorial unit of analysis a metropolitan region contributes to draw the attention of the relevance that the metropolitan agenda has for the country considering that the municipalities are interdependent both socioeconomically and environmentally and the importance of this level

for the formulation and planning instruments for territorial management. This study showed how the typology allows the identification of municipalities with great environmental heritage and social vulnerabilities, directing priority actions to these areas.

Once identified, the municipalities that are natural resource providers, but that have infrastructure deficits and greater inequities, are priorities for thinking about programs and direct actions that involve different levels of government and social actors. Elaborating a statistical instrument that allows the public power to visualize these areas can facilitate actions to reach out to municipalities that are more vulnerable. Lack of sanitation infrastructure (precarious sanitation, water management, and solid waste) is associated with degraded water supply areas resulting in poor quality of distributed water, exposing the population to water shortages and diseases.

Results from this study suggest that the metropolis management, particularly in regards to sustainability, must be conducted in an intermunicipal and integrated approach. Disregarding the heterogeneity of conditions in each locality of the metropolitan system may result in the failure and unsustainability of public policies when cities and its surrounding municipalities are considered in isolation.

The typology approach used in this study provides support to encourage the formation of cooperative political arrangements between municipalities. Clusters formed by their similarities share common issues and deficits, constituting areas with potential for the organization of intermunicipal cooperation groupings.

In Brazil, the Constitution regulates and provides for the formation of cooperative instruments via the municipal public consortia regulated by Federal Law Nº. 11.107/2005 [44]. Cooperation between the federative units is an alternative to promoting the development and quality of municipalities in order to solve problems and obtain joint results of a nature superior to the individual political, financial, and operational capacities of the municipalities, without losing the capacity of local governments to make their own decisions.

This association could allow better provision of social services to the entire population and preserve natural resources. However, this study has seen that as municipal consortium systems are a possibility for the formulation and management of public policies at the regional level, there are still limits to the efficiency of these arrangements. In practice, only interested municipalities adhere to intermunicipal management for some specific urban issues (e.g., solid waste management, sanitation, and water sources). Another challenge is the discontinuity of the programs and political arrangements with public management changes.

The methodological approach of this work is an innovation in terms of thinking about a metropolitan region in an integrated, decentralized way. It prioritizes the environmental dimension as the fundamental guide of urban planning to provide quality of life of its inhabitants beyond economic development. Few academic studies focus on the regional territory as a planning unit [9]. Generally, studies are more local and centralized in the administrative limits of the municipal autonomy.

Integrated regional approaches are increasingly fundamental to encompass the question of the sustainability of large metropolitan regions and their adjacent areas of influence, as they are beyond the domain of isolated administrative capacity of municipalities, necessitating regional and global political articulations.

## 5. Conclusions

In conclusion, the typology methodology used in this study has demonstrated the importance of identifying similarities and disparities in terms of social and environmental health conditions in a metropolitan area such as São Paulo, considering local characteristics from a sustainability perspective.

In addition, the use of multivariate analysis to group MRSP municipalities according to the DPSEEA Matrix showed to be a promising strategy for territorial analytical typology in environmental health. In this study, four groupings of municipalities that varied in relation to environmental and social conditions were identified for the MRSP. Grouping 1 presented higher values or deficiency of sewage

infrastructure and water supply and in contrast to these urban deficits but also a large percentage of native vegetation and watershed areas. Grouping 2 presented municipalities with better values for sanitation infrastructure coverage; however, it is where the largest social inequalities of access of the MRSP are concentrated. Grouping 3 was the most heterogeneous of all, containing municipalities with distinct characteristics, having similarities with Grouping 1 in environmental terms, with a large percentage of native vegetation. Groupings 1 and 3 should therefore be a priority in regional planning considering their role in providing ecosystem services essential for the environmental sustainability of the territory. Finally, Grouping 4, the municipality of São Paulo, showed the concentration of the greatest wealth and infrastructure but also MRSP's major social inequalities and important forested remnants and watershed areas. This study demonstrated that acknowledging the strength, weaknesses, and interdependence of these factors is essential for the sustainability of a megacity such as São Paulo, considering its economic importance but also its responsibility on the pressures that its expansion generates on socioeconomic discrepancies and environmental degradation.

The formulation of a typology in environmental health using a matrix of indicators as an aid to visualize the socioenvironmental conditions found in the metropolitan territory highlights the importance of the regional scale as a new territorial entity. Not only can the typology support the identification of instruments and solutions, which are better suited to deal with the challenges and demands of contemporary urban problems, but it also offers the potential to be replicated to other metropolitan regions in the world facing similar issues. Studies focusing on integrated environmental problems in a metropolitan scale offer the possibility to tackle contemporary challenges of megacities, particularly when related to the uneven use of resources.

Finally, extending the use of the typology methodology, aggregated by statistical tools, would offer the potential to integrate Sustainable Development Objectives in regional planning [10], not only at the Metropolitan level but to new territorial urban dimensions such as the Macrometropoles, offering potential for further research.

**Author Contributions:** N.C.M. developed this research during her master's project defining all the applied methodology participating in all phases from data collection, statistics degeneration to writing, review and editing of this article; A.R.M.-S. assisted in the application of statistical analysis, writing, proofreading and text editing; A.D.S. mentored and participated in the writing, proofreading and editing of the text.

**Funding:** CAPES Foundation (Coordination of Improvement of Higher Level Personnel)—Ministry of Education, Brazil, and Graduate Program in Public Health for the financial support for the publication.

**Conflicts of Interest:** The authors declare no conflict of interest.

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
