# Peer review of "An Environmental Health Typology as a Contributor to Sustainable Regional Urban Planning: The Case of the Metropolitan Region of São Paulo (MRSP)"

_sustainability, doi:10.3390/su11205800_

Round 1

Reviewer 1 Report

Thematic is of interest and is current considering the difference in development models that occur in metropolitan areas.
The search for identification of determinant elements that can help in the sustainable urban planning process is today one of the paths pursued by many societies and scientific communities. It follows from my position that the manuscript needs revision on the following points:
1 - Deepening the literature review requires updating based on international scientific publications and reference on the subject;
2 - The context that the current metropolitan scenario requires identifying which instruments already exist today for application to regional and supra-municipal areas that may be the starting point for a future process of indicator selection;
3 - It is necessary to make clear in the text what the authors consider as "... possible integrated analyzes of sustainable urban planning ...";
4 - The proposed research methodology should be more detailed so as to enable other researchers to understand its application and be able to replicate it to other contexts;
5 - Need to deepen the choice and justify how to select the indicators applied in research;
6. Conclusions should focus more on findigns than expectations and future developments.
I assume that the text should be revised.

Author Response

Please see the attachment,

Thanks

Reviewer 2 Report

Comments to the Authors

The paper titled “An Environmental Health Typology as a Contributor to Sustainable Regional Urban Planning: the Case of the Metropolitan Region of Sao Paulo (mrsp)” presents an interesting point of view on the methodology to interpret the broader term of “Sustainability” in complex urban environments, especially in large-scale urban areas.

The paper is coherently organized, the title is self-explanatory, a small advice is to rewording the aim of the paper in abstract, to make the core message clear from the beginning.

Overall, the entire paper is an interesting, original and coherent product of research, with minor issues related to minor misspelt words and some inconsistencies in the discussion.

As an example, in Row 4, the acronym is not in capital letters, as happens in Row 24.

I suggest to rephrase the statement between Row 70 and 74, adding also some example and reference of the kind of disparities (not only GDP but probably also access to infrastructures, lack of proper schools for low-income population and so on, if possible with a proper bibliography in order to better understand what kind of discrepancies we are facing with).

At the end of paragraph 1. Introduction I suggest to add a scheme of the paper, with the description of every paragraph, this can help to better understand the flow of the paper.

Paragraph 2.Materials and Methods is well described but can be useful add the proper reference and the date of updating of all datasets to check the coherence (i.e. UNEP – what document? When it was developed?).

In the phrase between Row 152 and 155 the categories of indicators can be related to a reference or, at the opposite, can be useful a note in which you explain why you decided to choose those indicators.

Can be useful, in Rows 309, 347, 384 and 429, giving a name to each type, to better understand what do you think is important to underline. The tables use the word “Grouping” instead of “Type”, there is a reason for this? Otherwise, my advice is to maintain just one name for every category, to avoid misunderstanding.

I agree with the authors’ statement between Row 497 and 500, but can be useful to add, in the introduction a clear statement of what the authors mean with “Sustainability”, and explaining why the economic indicators are voluntarily avoided in the model.

In the conclusions, I suggest answering the following questions: Can your method be used by a general readership in other areas of Brazil or Latin America? What are the effects of your method for the management of the metropolitan area?

Finally, I found the paper extremely well done, supported by an appropriate research method and discussed clearly and coherently, also if there is some minor problem of intelligibility due to a different approach to the broader context of sustainability.

Discussion, Conclusions and Bibliography are appropriated and balanced. Overall, the paper is clear and easy to understand. The results are interesting with good potential for future researches.

Hope my comments will be useful to the authors.

Author Response

Please see the attachment, 

Thanks

Round 2

Reviewer 1 Report

Many thanks for the improvement of yours text.

Author Response

We would like to thank you for all the contributions to our research and we resubmit the article seeking to answer all the questions raised. Thank you for the opportunity and for your contributions to our study.  
